# Emerging Nanoparticle Strategies for Modulating Tumor-Associated Macrophage Polarization

**DOI:** 10.3390/biom11121912

**Published:** 2021-12-20

**Authors:** Lu Shi, Hongchen Gu

**Affiliations:** Nano Biomedical Research Center, School of Biomedical Engineering & Med-X Research Institute, Shanghai Jiao Tong University, 1954 Huashan Road, Shanghai 200030, China; shilu2018@sjtu.edu.cn

**Keywords:** macrophage polarization, tumor-associated macrophages, nanoparticles, drug delivery, tumor microenvironment, cancer immunotherapy

## Abstract

Immunotherapy has made great progress in recent years, yet the efficacy of solid tumors remains far less than expected. One of the main hurdles is to overcome the immune-suppressive tumor microenvironment (TME). Among all cells in TME, tumor-associated macrophages (TAMs) play pivotal roles because of their abundance, multifaceted interactions to adaptive and host immune systems, as well as their context-dependent plasticity. Underlying the highly plastic characteristic, lots of research interests are focused on repolarizing TAMs from M2-like pro-tumor phenotype towards M1-like antitumoral ones. Nanotechnology offers great opportunities for targeting and modulating TAM polarization to mount the therapeutic efficacy in cancer immunotherapy. Here, this mini-review highlights those emerging nano-approaches for TAM repolarization in the last three years.

## 1. Introduction

In the past decade, immunotherapy has shown great therapeutic efficacy in treating several kinds of malignancies, especially in hematological malignancy, lymphomas and skin cancer. For example, the complete remission (CR) rates of acute lymphoblastic leukemia can reach as high as 90% with chimeric antigen receptor (CAR)-modified T cell therapies [1]. Nevertheless, moderate patients can benefit from the current immunotherapy, especially in some solid tumors, such as intraepithelial neoplasia, with lower mutation, such as gastrointestinal tumors, breast cancer and ovarian cancer [2,3]. Indeed, 50–75% of patients have no response overall despite the furthest advance of robust immune checkpoint inhibitors (ICIs) [4].

The immune-suppressive tumor microenvironment (TME) has been considered as one of the main hurdles in solid-tumor immunotherapy [5,6,7,8]. In the TME, cells represent a dynamic network where cancer cells and various immune cells interact with each other and modulate tumorigenesis and development [9]. Of the immune cells in the TME, tumor-associated macrophages (TAMs) are the most abundant (up to 50% in some solid tumors) [10,11]. Macrophages are highly heterogeneous cells and are primarily categorized into two subtypes according to various cues in the context: anti-tumor M1 and pro-tumor M2. The biomarkers of M1 macrophages usually include CD80, CD86 and MHC II [12,13]. Interferon-γ (IFN-γ) and lipopolysaccharide (LPS) are usually leveraged to polarize macrophages into activated M1 both in vitro and in vivo [14,15]. The skewed-to-M1 macrophages usually generate a high level of inducible nitric oxide synthase (iNOS) and tumor necrosis factor-alpha (TNF-α), inclining to kill tumor cells [16]. M2 express CD68, CD163, CD206 and CD204 [12,16]. IL-4, IL-10, IL-13 are typical lymphokines for inducing M2 subtype in vitro [16]. M2 macrophages generally have low expression of IL-12 and produce a high level of IL-10 and arginase 1 (ARG-1) [17]. Tumor-associated macrophages (TAMs) are progressively educated by the TME and are usually considered as M2-like macrophages to support tumor growth [18,19].

There are a bunch of strategies targeting macrophages for tumor immunotherapy since the augment of TAMs often represents poor prognosis [10,11,20,21]. Emerging interests are focused on the polarization approach for macrophage’s highly plasticity, an important role in maintaining immune homeostasis and potential T-cell-activation effect as professional antigen-presenting cells. To realize macrophage polarization towards the M1 pole, several kinds of therapeutic agents are applied by promoting the M1-associated signal pathway and/or suppressing M2 polarization. Specifically, agonists of Toll-like receptors (TLRs) are notable examples for arousing M1 macrophage activation since TLRs are one kind of pattern recognition receptor in an innate immune system. Some TLR agonists, such as poly I:C, MO-2055 and R848, have been used in both pre-clinical [22,23,24,25] and clinical trials [10,26]. Cytokines and growth factors are more complicated, for they trigger M1-direction polarization and inhibit M2 signals simultaneously. Several kinds of micro RNAs (miR) can also affect the phenotype of macrophages by regulating gene expression [27,28], such as miR-155, miR-127 and miR-125b [28]. Despite the efficacy, systematic administration of these agents might arouse toxicity and serious side effects. To further reduce the unexpected risks and promote the efficiency of targeting delivery, nanoparticles are leveraged for modulating TAM polarization.

TAM-targeting nanoparticles have gained great attention recently for their potential in solid-tumor immunotherapy. A nanoparticle is defined as a particle with submicron size in any dimension by the International Union of Pure and Applied Chemistry (IUPAC) [29]. Owing to this nanometer size and other properties, nanoparticles have several advantages in biomedical fields, including: (i) relatively high surface area to increase loading efficiency (compared to micro-scale particles) [30]; (ii) tunable parameters to achieve specific targeting, systemic toxicity reducing and fine-tuned application in diagnosis and treatment (compared to free drugs or other reagents) [31,32]; (iii) relatively stable structure to provide a shield for the cargos to prevent drugs from early degradation (compared to free macromolecules for therapeutic usage) [33,34]. With the development of cancer immunotherapy, nanoparticles are discovered to have more advantages and potentials for immunology application by targeting and regulating TAMs. As is known, larger particles are cleared by the mononuclear phagocyte system and the liver, while particles with a small size (less than 10 nm) are cleared via the kidney [35]. Lots of studies have provided evidence for large nanoparticles passively taken in by macrophages [36,37,38]. Therefore, in order to realize targeting TAMs, nanoparticles might be tailored to have a relatively larger size and pathogen-mimicking shape, which macrophages tend to capture and phagocytose. Active targeting strategies are also leveraged by modifying nanomedicines with various ligands of some receptors that only peritumoral macrophages possess, such as scavenger receptors. Common ligands include M2-peptides, mannose and folate [39,40,41]. According to the matrix, nanoplatforms can generally be categorized into polymeric nanoparticles, lipid-based nanocarriers, inorganic nanomaterials and others. A variety of reprogramming agents are used, as aforementioned, and loaded onto the nanocarriers to achieve polarization, yet some nanoparticles themselves have the ability to reprogram macrophages in degradation. In all, this mini-review will focus on the various TAM-reprogramming nanoparticle strategies or nanomedicines with polarization effects in the last three years according to their material matrix. Challenges and hurdles in this burgeoning field are also mentioned at the end.

## 2. Polymeric Nanoparticles and Macrophage Repolarization

Polymeric nanoparticles are one of the most significant nanoscale formulations that are linked by monomers with covalent bonds [42]. A polymer can be engineered to have various forms, such as polymeric micelles, polyplexes and solid particles. For biomedical usage, these polymeric nanomaterials have several outstanding advantages: (i) the relatively simple manufacture technique, usually self-assembly or emulsion fabrication process, providing possibilities for off-shelf nanomedicines [43,44]; (ii) functional groups on the surface to enable modification to confer specific targeting; (iii) relatively biodegradable and biocompatible, especially for natural polymer nanoparticles, such as chitosan, alginate and dextran. Some synthetic polymers, such as polylactide (PLA), are also biologically safe and have been approved for drug delivery and tissue engineering fields by the Food and Drug Administration (FDA) [35,44].

Studies are divided according to the polymer matrix and the agents the nanoparticles carry. As aforementioned, some nanoparticles have the capability to polarize macrophages themselves without reprogramming agents, and the effects are varied for different modifications and residues. Ann-Kathrin Fuchs et al. found that both carboxyl-modified polystyrene nanoparticles and amino-modified ones succeeded in suppressing macrophages from polarizing towards M2 by down regulating the expression of CD200R, CD163, as well as IL-10, without affecting the M1 markers [45]. Yen-Jang Huang’s group discovered that hydrophilic polyurethane nanoparticles themselves had surface-dependent immunosuppressive properties, preventing macrophages from M1 polarization by decreasing the production of TNF-α and IL-1β [46]. Carboxyl-based nanoparticles were more suppressive than the amino-modified ones [46]. The two studies suggested that the effect of nanoparticles on macrophage polarization does not only depend on functional groups but also on other properties, and the nanomedicines should be estimated as a whole. Recently, membrane-coating technology has been used widely in the biomedical field [47,48], and some cellular membranes could be special agents to influence macrophage polarization. For example, cellular membranes of natural killer cells (NKs) and THP1 macrophages were coated with poly(lactic-co-glycolic acid) (PLGA) nanoparticles, resulting in ten times higher IL-6 than that in the control group [49]. Likewise, macrophage membranes were also used to coat PLGA, which delivers iron oxide and TLR agonist R837 to potentiate immunotherapy [50]. C.G. Da Silvaa’s group used PLGA as a biodegradable matrix core to simultaneously deliver R848, poly (I:C) and MIP3α, leading to the inhibition of TC-1 growth in pre-clinical experiments [51].

TLR agonists are significant in macrophage repolarization. Christopher B. Rodell et al. showed that the TLR7/8 agonist, R848, was one of the most powerful molecules for polarizing macrophages in the M1 direction in vitro among 38 immunomodulatory agents reported in the literature [24]. β-cyclodextrin nanoparticles encapsulated with R848 succeeded in regulating TAMs and increased the growth inhibition efficacy of cancer cells in various models together with anti-PD-1 therapy [24]. A lignin nanoparticle was also a candidate to carry R848 and targeted CD206-expressing M2 with specific peptides modified on the surface [52]. In vitro, the nanomedicine successfully promoted M1 marker TNF-α almost twenty times more than the control and reduced the tumor burden in mice [52]. In another group, acetylated chondroitin sulfate protoporphyrin polymer was developed to deliver R837 [53]. Together with the other polymeric micelle loading with Dox, it suppressed 4T1 growth in mice [53]. The TLR9 agonist CpG has also been carried to modulate macrophages. Jutaek Nam’s lab [54] used cationic polyethyleneimine (PEI) to absorb CpG and neoantigen peptides to form a polyplex nano-vaccine. In draining lymph nodes (dLNs) of vaccinated mice, the amount of CD86+ M1 increased, and CD206+ M2 decreased [54], showing the capacity of nanovaccines to regulate macrophages. Moreover, TLR agonists and other agents also show great potential for TAM polarization. Plasmid DNA and messenger RNA (mRNA) are two therapeutic agent forms of genetic materials and are also leveraged to edit M1 regulators. For example, a novel lipid-coating polymer termed PQDEA was developed to form polyplexes with IL-12 plasmid and inhibited three tumor models in mice with only four doses [55]. Therefore, to further prompt macrophage repolarization, F. Zhang et al. discovered that polymers coated with M1-polarization-associated mRNA and modified with di-mannose could target the CD206 receptor of macrophages, increase M1 macrophages and suppress three tumor models (Figure 1) [56]. The M1-associated mRNA coded transcription factors IRF5 and IKKβ, which are downstream proteins in the IFN I pathway, thus promoting M1 skew [56]. Yudong Song et al. also modified their polymer with mannose to carry two short interfering RNAs (siRNAs), which block VEGF and PIGF. As these two factors mediate M2 polarization and monocyte recruitments, blocking them resulted in promoting IL-12 and IFN-γ in TME [57]. HA and miR125b are effective in promoting M1 polarization. HA-PEI nanoparticles loaded with miR125b were fabricated and succeeded in promoting TAM polarization towards the M1 type after being intraperitoneally injected [58]. Another article pointed out that the N-(2-hydroxypropyl) methacrylamide (HPMA)-copolymer nanocarrier could also target CD11b+ TAMs and regulate the TME in situ by inducing M1 polarization [59]. Some biophosphonates, such as zoledronic acid (ZOL), were also used to mediate TAM repolarization, and a pH-sensitive dendritic poly-lysine nanoparticle loaded with ZOL could release it once inside the TME [60]. For the interference of key chemokines, a shrinkable polymer carrying BLZ-945, a CSF1R inhibitor, to regulate TAM succeeded in reducing CD206+ M2 from 40% to 15%, while IL-12 and IFN-γ in TME increased to three times that of the control [61].

## 3. Lipid-Based Nanomaterials and Macrophage Repolarization

Lipid-based nanomaterials are well-known for their low immunogenicity, high biocompatibility in nature and being relatively easy to enlarge the manufacturing scale, thus serving as a proper candidate for biomedical application [62,63,64]. Liposomes and lipid nanoemulsions are two representatives for lipid-based nanoparticles and we outline the two nanoformulations separately.

Liposomes are vesicles with a sphere shape and consist of one or more lipid bilayers. They have been in the biomedical research spotlight for several decades because they have special advantages, including: (i) the biomimetic structure where the core of liposomes can encapsulate hydrophilic drugs while the lipid layers can entrap hydrophobic agents. This trait of liposomes might conquer the inconvenient loading of drugs with different dissolubility; (ii) a size of 40–150 nm, exhibiting the same as natural exosomes and thus serving as a biological nanocarrier in therapeutics [65,66]; (iii) extraordinary biocompatibility to achieve fewer side effects. Of note, liposomes encapsulated with doxorubicin (Doxil^®^) are the first nanomedicine approved by the FDA. Liposomal cisplatin has also been approved as a drug in pancreatic cancer by the European Agency for the Evaluation of Medicinal Products (EMEA) [67]. Moreover, the fast development of microfluidics in recent years facilitates the large-scale manufacture of liposomes and reduces differences from batches to batches [68].

Exertions have been made in leveraging liposomal vesicles to carry different agents to polarize TAMs, such as bisphosphonate, siRNA, cytokines, chemokines and TLR agonists. A report proved that ZOL-loaded liposomes succeeded in repolarizing cancer-educated macrophages towards a pro-inflammatory subtype with increasing expression of iNOS and TNF-α [69]. Another study showed that the pegylated liposomal nanoparticles (PLNs) loaded with alendronate could suppress tumor growth and increase progression-free survival in tumor-bearing mice. Interestingly, the PLNs themselves, on the contrary, empowered immunosuppression and impaired T-cell immunity [70]. As aforementioned, cues in TME polarize TAMs towards M2-like subtypes, and among them, hypoxia-induced factor 1-α (HIF-1α) are notorious. Therefore, a lipid nanocarrier with HIF-1α-blocking siRNA was developed and succeeded in promoting the secretion of TNF-α and IFN-γ in the TME [71]. The amount of CD169+ TAMs was also enhanced to 1.5-fold over the control group [71], demonstrating that the nanostrategy had regulated TAMs towards an immune-supportive direction. The polarization effect of TLR agonists on TAMs has greatly benefited from nanotechnology. A research group found that R848-encapsulated liposomes together with anti-EGFR antibody treatment could significantly suppress tumor growth in mice. Intriguingly, they found that M1 macrophages induced antibody-dependent cellular phagocytosis more strongly than M2 [72]. In addition, cytokine and chemokine therapies, which also benefited from nanocarriers, are candidates for reprogramming TAMs because of their powerful roles in tuning cellular signal pathways and phenotypes. In a recent report, one kind of clinical-approved liposome was used to deliver mRNA, which edited a bispecific antibody to neutralize CCL2 and CCL5 (Figure 2) [73]. The nanomedicine down regulated IL-10, ARG1, MRC1 and CD206 and suppressed liver cancer growth in mice together with PD-1 antibody treatment [73]. Anujan Ramesh’s lab constructed a liposome to carry BLZ945 and SHP099 (inhibitors of CSF1R and CD47-SIRPα signal pathway, respectively,) and the nanoparticle significantly down regulated the expression of CD206 in a macrophage cell line Raw264.7 from ~75% to ~10%, increasing the M1/M2 ratio by six times [74]. Recently, the same group constructed another liposome to deliver BLZ945 and another inhibitor Selumetinib to promote M1 polarization [75]. Some drugs available in the clinic might also have positive effects on programming macrophages. For instance, liposomes loaded with simvastatin and vorinostat also had positive impacts in down regulating CD206 and promoting CD86 expression [76]. To overcome the off-target effects of regulating compounds, another research group modified sialic acid to their liposomes and loaded with zoledronic acid because of the hig -expression of sialic acid receptors in TAMs [77].

Lipid nanoemulsions can also be fabricated with microfluidic or with ultrasonic devices. Similar to liposomes, nanoemulsions are sphere-shaped particles consisting of several kinds of lipids in a droplet and usually exhibit a low polydispersity. For the oil-in-water system, lipid emulsion can entrap hydrophobic drug molecules inside the oil droplet. Different TLR agonists might contribute to macrophage polarization to a different degree. A nanoemulsion carried with TLR7/8 agonists, R848, skewed more macrophages towards M1 directions than that loaded with R837 [78]. Together with vaccine treatment, this nanomedicine prolonged the survival time of animal models with melanoma and cervical cancer for it might revert “cold” tumors into the “hot” ones [78]. Special therapeutic agents have also been discovered to have the ability to convert macrophages. Ye Hui et al. [79] constructed a lipid nanoemulsion to carry an isoflavone gained from *Psoralea corylifolia* L., termed Neobavaisoflavone. This anti-cancer material was able to switch M2 macrophages to pro-inflammatory M1 in vivo [79].

## 4. Inorganic Nanoparticles and Macrophage Repolarization

Inorganic nanomaterials are defined as nanoparticles composed of inanimate maters and usually include a metal matrix, such as calcium, iron and gold, and nonmetal materials, such as silicon. The original properties confer inorganic nanoparticles lots of merits, including: (i) relatively stable for long time period conservation and for strict sterilization conditions compared to organic materials [80]; (ii) fine controllability for the structure and various shapes with a low polydispersity index [81]; (iii) the intrinsic physical properties for multipurpose applications, such as superparamagnetism, up-conversion luminescence and surface plasmon resonance [81,82]. In fact, some inorganic nanoparticles have been clinically approved. For instance, iron oxide-based nanomaterials, ferumoxytol (i.e., Feraheme^®^) and ferucarbotran (i.e., Resovist^®^) have been used to treat iron deficiency and complete magnetic resonance imaging (MRI), respectively [35,83]. In order to demonstrate inorganic nanomedicines in macrophage repolarization, we outline the approaches according to their intrinsic properties and the agents as well.

As aforementioned, some nanoparticles have the ability to reprogram macrophages themselves, and iron oxide nanomaterials are one of them. In 2016, Saeid Zanganeh et al. [84] found that the clinical-approved iron oxide nanoparticle ferumoxytol could inhibit tumor growth by polarizing macrophages in the M1 direction. Interestingly, it was later discovered that both clinical-approved iron oxide nanoparticles, feracarbotran and ferumoxytol, were able to induce macrophage autophagy and arouse inflammatory response through TLR4-mediated signaling and oxidative stress [85]. With the extracts help from targeting molecules and other reprogramming agents to TAMs, these self-service nanomedicines might further promote the efficacy and retardation of tumor development. Jiaojiao Zhao et al. found that ferumoxytol surface-functionalized with poly(I:C) could achieve macrophage activation and suppress malignant melanoma in mice [86]. Similarly, other iron oxide nanoparticles have also been reported to be effective in turning macrophages into anti-tumor subtypes [83,87,88,89,90,91]. Coating iron oxide with a cellular membrane expressing SIRPα from genetic-edited cells was proven to be an effective strategy. The iron-containing nanoparticles were magnetically oriented to aggregate in the TME, and with the blockade of the “do not eat me” signal, TAMs were repolarized towards M1, eliciting potent immune responses and suppressing both B16F10 and 4T1 cancer growth with T lymphocytes [92]. Super-paramagnetic iron oxide nanoparticles (SPIONs) with different charges were compared in a recent study [93]. SPIONs with positive or negative charges could skew macrophages towards M1-like phenotypes with a great promotion in TNF-α production [93]. Hollow iron oxide with PI3Kγ inhibitor payload and modified with mannose efficiently promoted NF-kB p65 expression and reprogrammed TAMs to M1 [94]. In an animal model with human breast cancer MDA-MB-231, the hollow nanomedicine inhibited tumor development [94]. Hyaluronic acid (HA) was carried by various iron oxide nanoparticles to regulate macrophages to promote the polarization effect [95,96,97]. Of note, the clinical observation is in accordance with the story above, where the number of iron-containing TAMs in patients with non-small cell lung cancer is usually associated with tumor regression [98]. Researchers further confirmed that the phenomenon resulted from TAM repolarizing into pro-inflammatory type after consuming iron-containing sub heme [98].

Furthermore, iron-relative nanoparticles, calcium ions and some catalytic nanoparticles have also been proven to regulate macrophage phenotypes. Calcium iron itself could boost pro-inflammatory cytokines production, such as IL-1β [99]. Therefore, nanoparticle medicine using calcium might have positive effects on M1 repolarization. Xiao-Yan He et al. delivered HA and IL-12-coded plasmid DNA with peptide-modified calcium carbonate and succeeded in multiple M1 markers increasing in J774A.1 cells [100]. Additionally, a nano-catalytic medicine was developed to target mitochondrial DNA (mtDNA) of cancer cells [101]. With the mtDNA oxidative damagecaused by nanoparticles, the remaining parts of nucleic acid escaped from tumor cells and then entered TAMs as damage-associated molecular patterns (DAMPs), which turned macrophages into M1 types [101]. IL-1β, TNF-α, IFN-β and IL-18 production reached as high as the positive control (i.e., LPS-treated group) after treatment. In addition, this novel strategy was verified to be effective on a PANC-1 tumor xenograft model [101].

Like other materials, inorganic nanoparticles can also be loaded with different reprogramming cargos. Photo-responsive up-conversion nanoparticles loaded with TAM-reprogramming agents achieved photo-related therapy and macrophage polarization simultaneously [102,103]. Ru nanoparticles loaded with BLZ945 also worked to decrease iNOS and CD206 expressions [104]. Non-mental materials, such as mesoporous silicon nanoparticles (MSN), are famous for their porous structure, which contributes to the delivery of more polarization-related agents in vivo and in vitro [101,105]. Leonard et al. also leveraged MSN to load albumin-paclitaxel, promoting M1 polarization [106]. Mesoporous silica loaded with siRNA to block monocarboxylate transporter-4 (MCT-4) also succeeded in preventing TAM from M2 polarization for MCT play a significant role in maintaining the acidic TME (Figure 3) [105]. Liming Bian and his colleagues coated MSN with up-conversion materials and loaded it with calcium regulators, which are released under near-infrared rays. Eventually, calcium levels in the cells increase or decrease, resulting in macrophage polarizing towards M1 or M2, respectively [103]. Gold nanoparticles linked with mucin-1 peptides succeeded in promoting M1 polarization with increasing cytokines, such as TNF-α, IL-6, IL-10, as well as IL-12 [107]. Li et al. rapped albumin and paclitaxel into gold nanorods, which suppressed M2 polarization and modulated the TME in tumor-bearing mice [108].

Manganese dioxide nanoparticles are also leveraged to deliver repolarization agents. Tsai-Te Lu’s team prepared a core-shell structure nanomedicine whose core was manganese dioxide and the shell contained PLGA and lipid and modified it with an SP94 peptide to target the TME [109]. The nanomedicine was proven to repolarize bone-marrow-derived macrophages, as indicated by the higher expression of M1 markers and down regulation of M2 markers [109]. Another team coated manganese dioxide nanoparticles with HA and achieved similar effects on macrophage repolarization [110]. Black phosphorus nanoparticles [111] and mesoporous Prussian blue [112] could also load HA with low-molecule weight to polarize M2. Linnan Yang and his colleagues prepared a nanocomposite, layered double hydroxide, to deliver miR155 [113]. Importantly, free miR155 failed to improve M1 markers while the nano-regulators could promote TNF-α, IL-12 and iNOS production [113]. Lv Chen’s group developed a copper sulfide nanomedicine modified with folic acid to carry CpG and docetaxel, resulting in the enhanced efficacy of phototherapy [114]. Ink-blue titanium dioxide nanoparticles modified with zwitterionic chitooligosaccharide (COS) was also used to enhance TAM reprogramming to M1 because COS has the ability to improve IL-2, TNF-α and IL-12 secretion [115]. Although the promising outcomes have revealed the feasibility of inorganic nanomaterials targeting TAM repolarization, additional animal tests need to be performed to verify the toxic side effects and the efficacy in real clinical situations.

## 5. Other Nanomaterials

With an extensive examination of the traditional nanoparticles, researchers developed lots of innovative nanomaterials to target TAM repolarization. Cholesterol-modified pullulan nanogels were reported to target macrophages in lymph nodes with subcutaneous injection [116]. When it carried long antigen peptides and CpG, the nanogel entered TAMs and activated macrophages with elevation in IFN-γ and IL-12 [117]. Chen’s group employed another gel where calcium carbonate inside could deliver an anti-CD47 antibody to the TME [118]. It was designed to spray the peritumoral tissue after surgical removal of the primary tumor to eliminate the remaining cancer cells, and it showed positive results in two models [118]. Another gel developed by Pengyu Guo et al. contained gold nanorods and iron oxide nanoparticles for thermal therapy and M1 polarization, respectively [119]. Exosomes from M1 were like natural liposomes that carry pro-inflammatory cytokines to promote macrophage M1 polarization with or without modification [120,121]. Graphene oxide (GO) was also able to polarize macrophages in a size-dependent way when large GO induced M1 polarization and promoted inflammatory reactions both in vivo and in vitro. Nanoparticles, such as gadolinium endohedral metallofullerenols (Gd@C_82_(OH)_22_), could activate macrophages through NLRP3 and TLRs-NF-kB pathways and thus prompted M1 polarization [122]. There are also several novel developments containing metal-organic framework nanoparticles to carry CpG [123], polymetformin-based nanoemulsion to deliver IL-12 plasmid DNA [124] and engineering exosome-mimic macrophages nanovesicles [125].

## 6. Conclusion and Perspective

The past decade has witnessed fast developments and achievements in cancer immunotherapy. However, patients with advanced solid tumors have a relatively low response to the available immunotherapy, and the tumor microenvironment remains one of the main obstacles. In the TME, positive feedback loops seem to be the main tune that prompts tumor development and metastasis. Cancer cells and the assistant cells secrete multiple chemokines and cytokines to recruit monocytes and macrophages and polarize them to M2-like TAMs. Hence these pro-tumor TAMs propagate the immune-suppressive responses by releasing specific cytokines that confer cancer cells’ benefits with extended survival time and more chance to metastasize. Given that macrophages might be a potential target to promote efficacy in immunotherapy, various approaches have been leveraged to cut off the loop by repolarizing macrophages towards an anti-tumor subtype. Recent nano-strategies to reprogram macrophages have been highlighted and divided into four groups, including polymer, lipid-based nanoparticles, inorganic and other materials. Cargos that are proven to have a repolarization effect mainly include TLR agonists, cytokines and growth factors and other agents, such as micro RNAs. With the nanomedicines administration, TAMs gradually repolarize to M1-like subtypes, resulting in turning the microenvironment into an immune-supportive one and suppressing tumor development (Figure 4). These nano-strategies are summarized in Table 1.

With promising outcomes in the pre-clinical stages, there are still mysterious mechanisms of nanomaterials that need to be disclosed further to testify the efficacy and solve safety concerns. In our points of view, four challenges should be conquered in this field: (i) the manufacturing process of nanoparticles can be scaled-up with neglectable differences in size, charges, surface-functional modifications and so on; (ii) the nanomedicines should be tailored to target macrophages with special locations or specific functions with minimum off-target side effects; (iii) the vehicles themselves should not have immunogenicity to avoid rejection if multiple dosing is needed; (iv) the optimal combination of programming drugs and nano-vectors. To overcome these obstacles, cutting-edge techniques are warranted in the nanomedicine field, including but not limited to: (i) microfluidic devices, they can be leveraged to generate nanoparticles with lower polydispersity and better reproducibility [68,135]; (ii) chips with abundant channels, which have gained a lot of attention in drug-screening applications [68,136] and can also be used to screen multiple ligand candidates to actively target TAMs simultaneously. Of note, the complex in vivo environment should also be noticed because the ligands of nanoparticles might be shielded by protein corona formed on the bio-nanointerface and affect the ultimate biodistribution [137,138,139]; (iii) organoid and tumoroid technologies. Recent advances in organoid and tumoroid might facilitate the evaluation of a new nanomedicine on the efficacy and safety in long-term toxicity [140]. Moreover, the toxicity that results from nanomaterials taken in by macrophages or other cells should also be taken into account in the beginning of designing a nanomedicine. All in all, the faster these emerging nanomaterials constructed from various novel biomaterials are developed, the more important the biocompatibility and biodegradability of the nanomedicine is; (iiii) Many computational algorithms and modeling approaches, such as geographical models, mathematical models and physical models, can be applied to simulate in vivo situations and provide significant insights for the rational design of a nanomedicine. For instance, Francesco Tavanti et al. [137] recently studied the interaction of common blood proteins with gold nanoparticles with Molecular Dynamics and the Martini coarse-grained model. They discovered that hydrophobic interactions played a vital role in protein binding to 11-mercapto-1-undecanesulfonate (MUS)-capped gold nanoparticles instead of electrostatic interactions, which dominate the process where proteins bind to citrate-capped gold nanoparticles [137]. In conclusion, these new techniques might provide tools for us to understand and predict the realistic behaviors of nanoparticles in biosystems.

With the significant role of macrophages in both innate and adaptive immune systems, nanomedicine specifically targeting TAM repolarization opens a new avenue in immune-regulated tumor eradication. The latest research further verified this with the fact that M1-like gene-edited macrophages managed to activate immature DCs and prompted T lymphocytes infiltration into the TME [141]. With all these discoveries of TAMs, we are now anticipating more clinical-translational contributions in this burgeoning field with TAM polarizing nanomedicines.

## Figures and Tables

**Figure 1 biomolecules-11-01912-f001:**
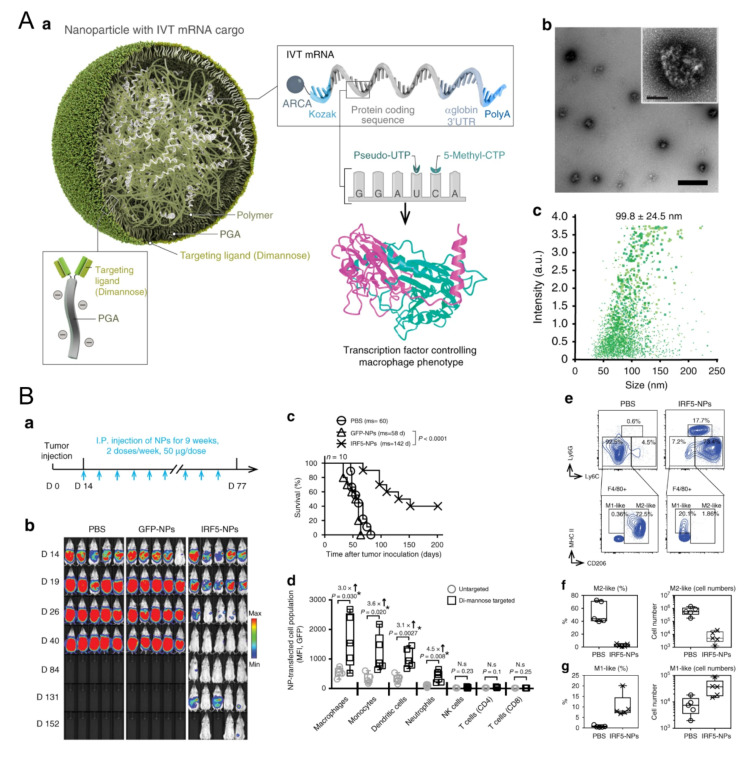
Polymeric nanoparticles loaded with mRNA for macrophage repolarization. (**A**) Characteristics of the polymeric nanocarriers: (**a**) The components of the polymeric nanoparticles (termed IRF5-NPs). (**b**) Transmission electron microscopy of the nanoparticles. (**c**) Size distribution of the nanocarriers. Reproduced with permission [56]. Copyright 2019 Nature Publishing Group. (**B**) IRF5-NPs prolong the survival time of mice with ovarian cancer and reprogram macrophages in vivo. (**a**) Treatment schedule. (**b**) Tumor growth after intraperitoneal administration of IRF5-NPs. (**c**) The survival curves of tumor-bearing mice. (**d**) Quantitation of transfection rates in various immune cells using flow cytometry. (**e**–**g**) The reprogramming effect of IRF5-NPs on peritoneal macrophages in tumor-bearing mice. Reproduced with permission [56]. Copyright 2019 Nature Publishing Group.

**Figure 2 biomolecules-11-01912-f002:**
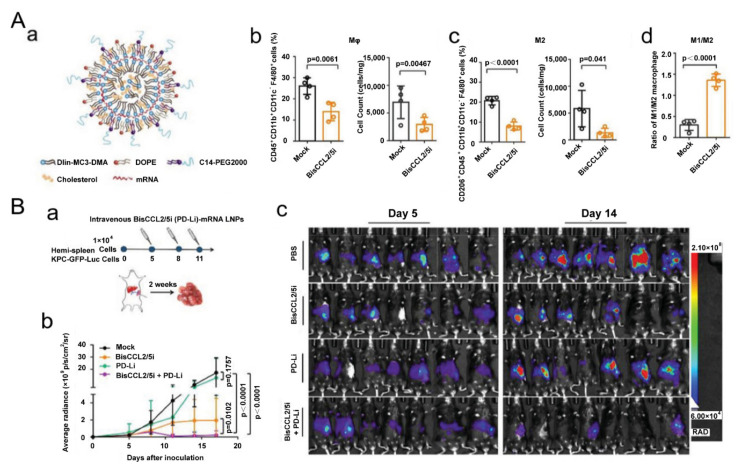
Lipid-based nanoparticles used for mRNA delivery of a bispecific single-domain antibody for repolarizing TAMs. (**A**) Lipid-based nanoparticles loaded with mRNA to realize CCL2 and CCL5 dual blockade reprogram TAMs. (**a**) Designs of the lipid-based nanocarriers. (**b**) The percentage of macrophages and (**c**) M2 subtypes in tumor tissue 48 h after systemic administration of the nanoparticles. (**d**) The ratio of M1/M2 in the TME. Reproduced with permission [73]. Copyright 2021 Wiley-VCH. (**B**) Dual blockade nano-strategy combined with PD-1/PD-L1 inhibition suppresses KPC liver cancer growth. (**a**) Time lines of the experiment. (**b**) Tumor growth after various treatments. (**c**) In vivo bioluminescence of tumor-bearing mice on days 5 and 14. Reproduced with permission [73]. Copyright 2021 Wiley-VCH.

**Figure 3 biomolecules-11-01912-f003:**
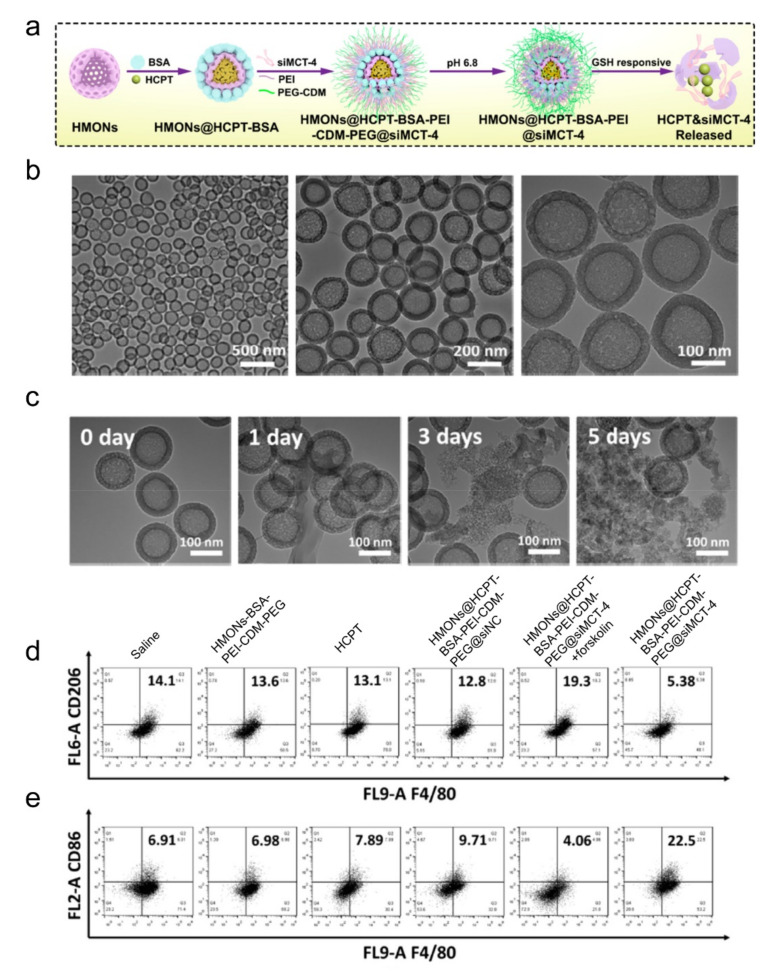
Inorganic nanoparticles with siRNA to block MCT-4 for repolarizing TAMs. (**a**) Schematic illustration for hollow mesoporous organosilica loaded with hydroxycamptothecin and siRNA-blocking MCT-4. (**b**) Transmission electron microscopy of the nanomedicines. (**c**) Transmission electron microscopy of the nanoparticles after incubating with a buffer containing redox glutathione for the number of indicated days. (**d**,**e**) Flow cytometric analysis of TAM phenotypes using the (**d**) M2 marker CD206 and (**e**) M1 marker CD86. Reproduced with permission [105]. Copyright 2020 American Chemical Society.

**Figure 4 biomolecules-11-01912-f004:**
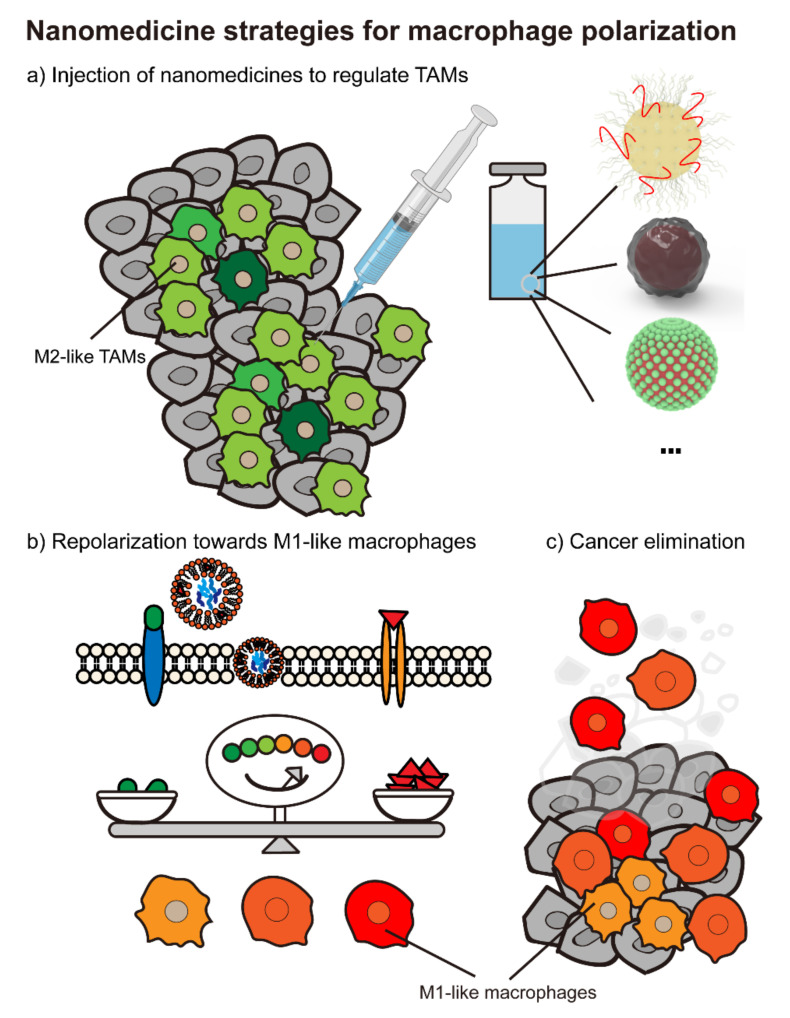
Nanomedicines targeting macrophage polarization. (**a**) There are plenty of nanoparticle strategies focusing on reprogramming M2-like tumor-associated macrophages into M1 poles, such as polymeric nanomaterials, lipid-based nanomedicines, inorganic nanoparticles and so on. (**b**) After the administration of nanocarriers, macrophages in the tumor microenvironment turn into a “friendly” subtype and (**c**) ultimately eliminate cancer cells together with other immune cells.

**Table 1 biomolecules-11-01912-t001:** Various nanostrategies targeting TAM repolarization.

Main Matrix	Therapeutic Agent	Tumor Model	Reference
Polymeric nanoparticles			
PLGA	Natural Killer cell membrane	4T1	[49]
PLGA	Iron oxide, M1 cell membrane	4T1	[50]
PLGA	CpG	B16F10	[126]
PLGA	R848, Poly (I:C)	TC-1	[51]
PGA	M-CSF	B16	[127]
PEI	Hyaluronic acid, miR125b	Nonsmall cell lung tumor	[58]
β-cyclodextrin	R848	MC38, B16F10	[24]
Lignin	R848	4T1	[52]
Poly (ε-caprolactone) (PCL), Sulfate protoporphyrin	R837	4T1	[53]
PQDEA	IL-12 plasmid	KPC, BPD6, 4T1	[55]
PEI	CpG	MC38, B16F10	[54]
PEI	Paclitaxel, CRISPR/Cas9-Cdk5	CT26, B16F10	[128]
Trimethyl chitosan	siRNA blocking VEGF, PIGF	4T1	[57]
Poly-L-lysines	Zoledronic acid	4T1	[60]
Poly (ethylene glycol) -b-PHEP (PEG-b-PHEP)	BLZ-945	4T1	[61]
PEI-PCL	Shikonin	CT26	[129]
PCL-Hyd-PEG	CpG, anti-CD80 antibody	4T1, B16F10	[130]
Polymetformin	IL-12 plasmid, hyaluronic acid	4T1	[124]
Nanomicelles named as QHMF	Hyaluronic acid	A549	[131]
Lipid-based nanoparticles			
Lipid nanoparticles	siRNA blocking STAT3, HIF-1α	OS-RC-2	[71]
Lipid nanoparticles	IMD-0354	Hepa1-6	[132]
Liposome	R848	WiDr	[72]
Liposome	Bispecific antibody (binds CCL2, CCL5)	HCC, KPC liver tumor model	[73]
Liposome	BLZ945, anti-CD206	4T1, B16F10	[74]
Liposome	Alendronate	TC-1	[70]
Liposome	BLZ-945, Selumetinib	4T1	[75]
Liposome	Zoledronic acid	S180	[77]
Lipid nanoemulsion	R848, R837	B16F10-OVA, TC-1	[78]
Lipid nanoemulsion	Neobavaisoflavone	A549	[79]
Inorganic nanoparticles			
Lanthanide-doped upconversion nanocrystals	Hyaluronic acid	/	[102]
Mesoporous silica with upconversion materials	Calcium ion	/	[103]
Iron oxide	Iron oxide, membrane blocking CD47-SIRPα	4T1, B16F10	[92]
Iron oxide	Iron oxide	/	[85]
Iron Oxide	Iron oxide	HT1080	[93]
Iron Oxide	Iron Oxide, 3-MA	MDA-MB-231	[94]
Iron Oxide	Iron oxide	4T1	[89]
Iron Oxide	Iron oxide	4T1	[90]
Iron Oxide	Iron oxide, hyaluronic acid	4T1	[96]
Iron Oxide	Iron oxide, hyaluronic acid	4T1	[97]
Iron Oxide	Iron oxide, poly (I:C)	B16F10	[86]
Iron Oxide	Iron oxide and hyaluronic acid stimulated macrophages	4T1	[95]
Berlin blue	Hyaluronic acid	4T1	[112]
Iron Oxide	Iron Oxide	E.G7-OVA	[91]
Rubidium	BLZ-945	CT26	[104]
Iron chelated nanoparticles	Iron	CT26, 4T1	[133]
Silica	Ferrous ion, rubidium ion	PANC-1	[101]
Silica	siRNA blocking MCT-4	4T1, B16F10	[105]
Calcium carbonate	Hyaluronic acid, IL-12 plasmid	/	[100]
Copper sulphide	CpG	4T1	[114]
Titanium dioxide	Chitooligosaccharide	H22	[115]
Black phosphorus	Hyaluronic acid	4T1	[111]
Layered double hydroxides	miR155	TC-1	[113]
Other nanoparticles			
Cholsterol pullulan nanogel	CpG	CMS5a	[117]
Fibrin gel (containing calcium carbonate)	Anti-CD47 antibody	B16F10	[118]
Gel containing iron oxide and gold nanorod	Iron oxide	MB49	[119]
Exosome	siRNA blocking galectin-9	PANC-2	[134]
Exosome	Exosome from M1	4T1	[120]
Metal-Organic framework	CpG	MDA-MB-231	[123]
Nanoparticles obtained from iron-oxide-stimulated macrophages	Iron oxide	4T1	[125]

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
