# Peer review of "Emerging Nanoparticle Strategies for Modulating Tumor-Associated Macrophage Polarization"

_biomolecules, 2021, doi:10.3390/biom11121912_

Round 1

Reviewer 1 Report

The current manuscript summarized the application of nanomedicine to modulate macrophage polarization. This is a fast developing area and the review regarding these studies is important. However, there are still some drawbacks with the current manuscript. Please find my comments below:

1, Significant amount of references are missing in some paragraphs including the paragraph starting with"TAM-targeting nanoparticles..." and "Polymeric nanoparticles are one of the most significant nanoscale..." and "Lipid-based nanomaterials are well-known for their low immunogenicity..." and "Inorganic nanomaterials are defined as nanoparticles composed of inanimate maters".

2, The manuscript fails to provide a in-depth discussion regarding the rationale behind the targeting mechanism of nano-particles to TAM.

3, In the perspective part, the authors summarized four challenges however they fail to provide any relevant or promising solutions to task those challenges. 

Author Response

Response to Reviewer 1 Comments

We want to begin by thanking Reviewer #1 for writing that” the review regarding these studies is important.” We also appreciated the constructive criticism and suggestion. We addressed all the points raised by the reviewer, as summarized below.

Point 1: Significant amount of references are missing in some paragraphs including the paragraph starting with"TAM-targeting nanoparticles..." and "Polymeric nanoparticles are one of the most significant nanoscale..." and "Lipid-based nanomaterials are well-known for their low immunogenicity..." and "Inorganic nanomaterials are defined as nanoparticles composed of inanimate maters".

Response 1:

The referee correctly noted that we missed some important references in some paragraphs.

Thanks to the referee’s reminder, we added several significant references in those paragraphs.

1)    References included ref. 30 in line 70, ref. 31,32 in line 72, ref. 33,34 in line 74 in paragraph starting with “TAM-targeting nanoparticles…”.

2)    In paragraph starting with “Polymeric nanoparticles are one of the most significant nanoscale…”, references newly added included ref. 42 in line 95 and ref. 43,44 in line 99.

3)    In paragraph starting with “Lipid-based nanomaterials are well-known for their low immunogenicity”, we added ref. 62,63,64 in line 179.

4)    In paragraph starting with “Inorganic nanomaterials are defined as nanoparticles composed of inanimate maters”, we added ref. 80 in line 258, ref. 81 in 259, ref.81, 82 in line 261 and ref. 35, 83 in line 264.

5)    With the kind reminder of the reviewer, we checked again our manuscript for any reference missing and we added ref. 105 in Figure 3 in line 351. Thanks again.

Point 2: The manuscript fails to provide a in-depth discussion regarding the rationale behind the targeting mechanism of nano-particles to TAM.

Response 2:

The reviewer pointed out that we failed to provide an in-depth discussion related to the targeting mechanism of nanoparticles to TAMs. We revised the introduction part by supplementing two ways for nanoparticles targeting TAMs:

 1) Larger nanoparticles passively target macrophages because macrophages tend to uptake them.

2) Nanoparticles with ligands modified can actively target TAMs since some receptors are only expressed by TAMs. We also provided some common ligands including M2-peptides, mannose and folate. They were demonstrated from line 76 to line 85.

Point 3: In the perspective part, the authors summarized four challenges however they fail to provide any relevant or promising solutions to task those challenges.

Response 3:

According to the reviewer’s suggestion, we supplemented four possible solutions to the four challenges in our perspective part from line 408 to the end of the paragraph. We thought cutting-edge techniques are warranted to task those challenges, including but not limited to microfluidics devices to generate nanoparticles with neglectable differences, chips with multiple channels to screen TAM-targeting ligand candidates, organoid and tumoroid to study the toxicity of a new nanomedicine and computation methods to simulate in vivo situation. 

Thanks again to the reviewer’s comment. We believe the additional changes we have made in response to the reviewer’s comments have made this a better manuscript.

Reviewer 2 Report

The review is well written and almost all results on TAM are described. I have one major suggestion: authors do not take into account the fate of nanoparticles after the drug release that can lead to toxicity. Moreover, authors should clearly state that nanoparticles undergo to some transformations upon interaction with biological medium, i.e. the protein corona formation, that can alter the fate of NPs. 

I suggest some papers that could be interesting for this topic:

https://doi.org/10.1039/C2AN35863H

https://doi.org/10.1002/adma.201703704 https://doi.org/10.3390/ijms22168722 In line 67, authors say that NPs have a size between 1-500nm. This is not completely true. Small NPs can be viewed as quantum dots. From line 133, the text is not formatted as the whole text.

Author Response

Response to Reviewer 2 Comments

We would like to thank the Reviewer#2 for writing that “the review is well written and almost all results on TAM are described”. We truly appreciate the thoughtful suggestions. Below is our point-by-point response to the reviewer’s comments.

Point 1: I have one major suggestion: authors do not take into account the fate of nanoparticles after the drug release that can lead to toxicity.

Response 1:

The reviewer correctly pointed out that we failed to consider nanoparticles’ fate in biological system.

1) We first corrected that the toxicity reducing effect of nanocarriers are actually systemic toxicity in line 71.
2) We then added “Moreover, the toxicity that results from nanomaterials taken in by macrophages or other cells should also be taken into account in the beginning of designing a nanomedicine.” in perspective part from line 418 to 420.

Point 2: Moreover, authors should clearly state that nanoparticles undergo to some transformations upon interaction with biological medium, i.e. the protein corona formation, that can alter the fate of NPs.

I suggest some papers that could be interesting for this topic:

https://doi.org/10.1039/C2AN35863H

https://doi.org/10.1002/adma.201703704

https://doi.org/10.3390/ijms22168722

Response 2:

According to the reviewer’s suggestion, we carefully read the two articles and one review kindly provided by the review and we then supplemented some opinions regarding the bio-nano interface and biodistribution of nanomedicines.

1)We pointed out from line 413 to line 415 that the design of nanomedicine should take the complex in vivo environment into consideration because the ligands of nanoparticles might be shielded by protein corona.

2)We also recommended computational methods to study nanoparticles’ behavior in biosystems from line 422 to line 430. References were supplemented as well to elucidate the situation.

Point 3: In line 67, authors say that NPs have a size between 1-500nm. This is not completely true. Small NPs can be viewed as quantum dots.

Response 3:

Thanks to the reviewer for their careful reading, we revised our incorrect definition of nanoparticles in line 67 and used the definition from IUPAC.

Point 4: From line 133, the text is not formatted as the whole text.

Response 4:

Thank to reviewer’s reminder, the wrong format from line 143 (line 133 in former version) was corrected.

Thanks again to the reviewer’s comment. We believe that the changes and revision we have made in response to the reviewer’s comments have made this a much clearer and broader manuscript.

Round 2

Reviewer 1 Report

No.

Response:

Thanks.